# Corrosion of NiTiDiscs in Different Seawater Environments

**DOI:** 10.3390/ma15082841

**Published:** 2022-04-13

**Authors:** Jelena Pješčić-Šćepanović, Gyöngyi Vastag, Špiro Ivošević, Nataša Kovač, Rebeka Rudolf

**Affiliations:** 1Faculty of Metallurgy and Technology, University of Montenegro, Dzordza Vasingtona bb, 81000 Podgorica, Montenegro; 2Faculty of Sciences, University of Novi Sad, Trg Dositeja Obradovića 3, 21 000 Novi Sad, Serbia; djendji.vastag@dh.uns.ac.rs; 3Faculty of Maritime Studies Kotor, University of Montenegro, Put I Bokeljske Brigade 44, 85330 Kotor, Montenegro; spiroi@ucg.ac.me; 4Faculty of Applied Sciences, University of Donja Gorica, Oktoih 1, Donja Gorica, 81000 Podgorica, Montenegro; natasa.kovac@udg.edu.me; 5Faculty of Mechanical Engineering, University of Maribor, Smetanova ul. 17, 2000 Maribor, Slovenia; rebeka.rudolf@um.si

**Keywords:** NiTi discs, corrosion rate, real seawater environment, cluster analysis (CA), principal component analysis (PCA)

## Abstract

This paper gives an approach to the corrosion resistance analysis and changes in the chemical composition of anNiTi alloy in the shape of a disc, depending on different real seawater environments. The NiTi discs were analysed after 6 months of exposure in real seawater environments: the atmosphere, a tidal zone, and seawater. The corrosion tests showed that the highest corrosion rate for the discs is in seawater because this had the highest value of current density, and the initial disc had the most negative potential. Measuring the chemical composition of the discs using inductively coupled plasma and X-ray fluorescence before the experiment and semiquantitative analysis after the experiment showed the chemical composition after 6 months of exposure. Furthermore, the applied principal component analysis and cluster analysis revealed the influence of the different environments on the changes in the chemical composition of the discs. Cluster analysis detected small differences between the similar corrosive influences of the analysed types of environments during the period of exposure. The obtained results confirm that PCA can detect subtle quantitative differences among the corrosive influences of the types of marine environments, although the examined corrosive influences are quite similar. The applied chemometric methods (CA and PCA) are, therefore, sensitive enough to register the existence of slight differences among corrosive environmental influences on the analysed NiTi SMA.

## 1. Introduction

In order to find better technical and commercial applications of materials in different industrial fields, the first smart alloy, or shape memory alloy (SMA), was discovered by Arne Ölander in 1932 [1]. The term “shape-memory effect” was recognised and described by William Buehler and Frederick Wang in a nickel–titanium alloy (an NiTi alloy callednitinol) in 1962 [2,3]. Other thermo-mechanical characteristics of shape memory materials, such as pseudoelasticity or high damping, were discovered later. Since then, different metallic alloys with shape memory behaviour and other inherent properties (physical, mechanical, and electrical) have been used in different industrial applications. Until now, different engineering and technical applications of SMA have been used for commercial fields, such as biomedical, automotive, aerospace, structures and composites, robotics, and even fashion [4]. Many industrial applications, such as different fluid connectors, couplings, valves, actuators, and others, can take advantage of the SMA effect. Furthermore, in maritime applications, SMAs can be used from the deep sea (for connecting tubes), under seawater (different manned or unmanned vehicles), on the sea’s surface (thermostats for transferring thermal energy in electricity under a power plant), or fixed marine structures (offshore oil extraction or a windfarm) [5].

Between a lot of alloy families based on Au, Al, Cu, Ni, Ti, etc., different binary or ternary groups of alloys were used in specific industrial fields. Until now, among the many different alloy families, the greatest shape memory effect has been exhibited by an alloy based on NiTi, Cu, and Fe. If we compare these three alloy families, we can conclude that NiTi has the highest recoverable strain, up to 8%. The fabrication costs of NiTi are high, while for the other two families of alloys, these are lower. Processing of NiTi alloy is technically complex, while Cu- and Fe-based alloy production is simpler and slightly less demanding [6]. Although Fe-based and Cu-based SMAs are low-cost and commercially available, due to their instability, impracticability (e.g., brittleness), and poor thermo-mechanic performance, NiTi-based SMAs are much more preferable for most applications [3]. The advantages of NiTi alloys are in their application at temperatures from −100 °C to 100 °C, hysteresis up to 30 (°C), and maximum recovery strain up to 8% [7].

One of the most important characteristics of SMAs on which the thermo-mechanical characteristics of the alloy depends is the chemical composition and grain size. Furthermore, different experimental and numerical analyses can be conducted on the mechanical properties of shape memory materials. In order to achieve better characteristics of alloys, various production processes can be applied, such as induction melting, laser melting, electric arc melting, the melt spinning technique, powder metallurgy, combustion synthesis, etc. [8,9]. This is followed by procedures of the thermo-mechanical heat treatment (forging, rolling wire drawing, rolling), which, finally, lead to the desired properties of the alloys.

When considering the production of NiTi alloys, casting processes, vacuum induction melting, vacuum arc remelting, electron beam melting, plasma arc melting, and electron beam melting are the most commonly used processes [10].Although expensive from the point of view of commercial application, NiTi alloys are the most important SMAs that have excellent mechanical properties (high ductility and low modulus of elasticity) and physical properties (wear resistance and biocompatibility). Their commercial application in different industries depends primarily on specific thermo-mechanical characteristics, such as shape memory effect, super-elasticity (mechanically induced), and one-way and two-way memory shape effect (thermally induced) [11,12].

The hardness and chemical composition of NiTi alloys are important for abrasion resistance, while toughness and hardening affect wear resistance under shocks or high stresses. However, it seems that these mechanical properties are not the main factors influencing the high corrosion resistance of NiTi. Previous research has shown that a significant influence can be attributed to the pseudoelasticity and chemical composition of the alloy. Different groups of NiTi alloys, with their combined properties of marine corrosion resistance, wear resistance, high hardness and strength, low density, low modulus of elasticity, and non-magnetism are imposed as optimal solutions in the maritime industry [12,13].

If we consider SMA applications, it can be concluded that using SMAs in marine applications requires huge research into the behaviour of this alloy in different laboratory or seawater environments. The complexity of the marine environment in different commercial SMA applications can be manifested through various physical phenomena of corrosion, such as general, intergranular, pitting, galvanic, crevice, stress, cavitation corrosion, corrosive fatigue, etc. [14]. According to the available research, it was discovered that the NiTi alloy has good resistance to stress corrosion cracking and good resistance in the marine environment [15]. However, corrosion tests have shown that impact corrosion, cavitation corrosion, stress corrosion [16], and pitting corrosion [17] can occur in different marine applications.

Considering contemporary development trends in the area of developing SMAs, the following have been found in some specific directions: improved properties in existing SMAs, fabrication of new SMAs, improved recovery stress, optimised alloy composition, or determination of the corrosion rate [16]. The application of SMAs in various industries entails the inclusion of specific environmental conditions; thus, the application of SMAs in the maritime industry requires an understanding and analysis of the impact of the complex conditions of the marine and coastal environments.

Due to corrosion resistance, high-hardness and high-strength nitinol can be used in different hydrospace applications as they can be non-magnetic hand tools or bearings for water-flooded rotating components [12]. Furthermore, shape memory alloy has been used in pipe line couplings for shipbuilding or as a connection for tubes for coupling in subsea environments, while nitinol wire can be used for SMA thermostats [18,19,20,21].

In previous research [22,23,24,25], the authors analysed the corrosion rate of an NiTi alloy obtained by the focus ion beam method. In this study, we analysed changing corrosion potential as a function of time, polarisation resistance, and the potentiodynamic method. In addition, with the extensive database on the chemical composition of alloys obtained on the basis of EDX analysis, we applied the CA and PCA methods to consider whether changes in the chemical composition of alloys could be observed, depending on the environment in which the alloys are located.

## 2. Materials and Methods

### 2.1. Materials

In this research, pure metals were used to produce NiTi: Ni (99.99 wt.%) and Ti (99.99 wt.%) delivered by Zlatarna Celje d.o.o. Slovenia. An alloy of NiTi with 50 at.% Ni and 50 at.% Ti was produced through classical casting and rolling. The Ni and Ti components (5.5kg Ni and 4.5 kg Ti) were weighed according to the Ni-Ti phase diagram, where the required Ni-Ti ratio was 50/50 at.% and 55/45 wt.%, respectively, to achieve an NiTi alloy with a microstructure having shape memory properties. NiTi alloy was manufactured through the classical process, which consisted of remelting in a vacuum furnace (*p* = 10 mbar), due to the great tendency for the oxidation of Ti (−2.3 kJ/mol), with electro-resistive heating at T = 1450 °C, until all the components were well-mixed. The final casting was formed into an ingot with a diameter of 65 mm, wherein the casting was cooled slowly to room temperature. The NiTi cast was then subjected to a mechanical treatment, profile rolling, with intermediate annealing at T = 1000 °C so that a reduction in diameter of up to 43 mm was achieved. Discs with a diameter of 42.3 mm and thickness of 3.4 mm and a hole were made for testing with the electro-erosion cutting process. Due to contamination when cutting with brass wire, all discs were specially polished by hand to avoid the presence of Cu or Zn on the surface. The hole was designed to clamp the plastic flax, so that the discs could be positioned at selected locations in the seawater environment (Figure 1a).

#### 2.1.1. Preparation of NiTi Discs for Microstructure Observation

Prior to performing the corrosion tests, the discs were subjected to metallographic preparation in order to detect their initial microstructure, which allowed comparison with the microstructure after the performed corrosion tests. Samples from the discs were placed in a hot investment mass, so that it was possible to grind the surface with abrasive paper of 180-4000 # on the grinding and polishing machines, Buhler Automet 250 and Eco Met 250 (Buhler, Swiss). Polishing with felt and polishing with an Al_2_O_3_ suspension of 1 μm followed on the same devices. After polishing, the discs were cleanedusingultrasound. This was followed by mild chemical etching with a solution of 3 mL HF, 6 mL HNO_3_, and 100 mL H_2_; etching time was 105 s [23].

The Nikon Epiphot 300 (Japan) optical microscope was used for observations of the initial output microstructure. Figure 1b shows the representative optical microstructures of the NiTi discs. The microstructure is typically a lamellar eutectic structure with phase composition: NiTi and Ni_3_Ti, which is consistent with previous research [26]. The bright phase is NiTi, containing around Ni = 50 at.% and Ti = 50 at.%, while the second darkminority phase is a eutectic phase composed of NiTi_eut._ and Ni_3_Ti_eut._ (Ni = 75 at.% and Ti = 25 at.%). The grains were almost the same size (about 30 μm), and the grain boundaries were clearly noticeable.

#### 2.1.2. Chemical Composition of the NiTiDiscs

Using inductively coupled plasma (ICP) analysis (performed to identify and measure the range of chemical elements) and X-ray fluorescence (XRF) composition analysis, the compositions of the NiTi discs were obtained [23].

#### 2.1.3. Microstructure Observation

A scanning electron microscope (SEM), Sirion 400NC (FEI, Hillsboro, OR, USA), with an energy dispersive X-ray spectroscope (EDX) INCA 350 (Oxford Instruments, Oxfordshire, UK) was used for the detailed microchemical analyses of the NiTi discs.

After seawater testing, the NiTi discs were inserted into the microscope chamber without prior preparation in order to preserve authenticity.

### 2.2. Proposed Problem and Related Methodology

This research covers the analysed behaviour of NiTi discs after 6 months of exposure in three different seawater environments in two specific directions. The first direction is to determine the values of corrosion potential and current density for each of the tested discs. In the second direction, using semiquantitative analysis, we obtained the chemical composition of the discs’ surfaces and applied principal component analysis and cluster analysis in order to discover the influence of the different environments on the change in the chemical compositions of the discs.

#### 2.2.1. Corrosion Measurement Method

Corrosion and electrochemical tests were performed on Princeton Applied Research accelerated test equipment consisting of apotentiostatandgalvanostat model 273, differential electrometer, corrosion cell K0047, standard saturated calomel electrode, auxiliary electrodesroller electrographite, computer with corrosion SOFTCORR 352 II software and a printer [27]. The tested discs were prepared mechanically on 2000 grit sandpaper. The corrosion investigations were performed using the following methods: change incorrosion potential as a function of time, linear polarisation, and the potentiodynamic (Tafel) method, in a corrosive medium such as an aqueous 3% NaCl solution.

##### Change in Corrosion Potential as a Function of Time

Corrosion processes can be monitored by changing the corrosion potential over time for defined operating conditions and are used successfully to interpret corrosion phenomena [22]. Three types of diagrams are usually obtained in such tests:

The first type of diagram indicates the activation of the metal surface. The electrode potential shifts to more negative values over time, and the metal becomes less noble. In this case, the metal dissolves faster and the corrosion rate increases over time.

A Type II diagram indicates the passivation of the metal’s surface. The potential becomes more positive over time, and the metal becomes nobler in relation to the initial state. The passive layer on the surface has an increasing thickness, as it becomes more compact, partially or completely non-porous, so that it prevents the diffusion of ions to the metal surface, thus slowing down the corrosion process.

A Type III diagram is characterised by the fact that the potential does not change over time, so further tests reveal whether the metal behaves constantly corrosively active or passive.

##### Linear Polarisation Model

This is an accurate and very sensitive method, which serves to determine the absolute degree of corrosion of metals and alloys and requires the supply of an external current source. This method produces values of the following elements: corrosion current density, polarisation resistance, and potential at a current value equal to zero, i.e., E (j = 0).

With this method, it is important to define the scan rate, i.e., the speed of voltage supply, which ranges from 0.1 to 10 mV/s. A scanning speed of 1 mV/s is most commonly used [27].

##### Potentiodynamic (Tafel) Method

This method records potentiodynamic curves. The E (j = 0), cathode, and anode Tafel constants are determined based on them. As with the previous methods, the chemical composition and structural composition of the disc, as well as the standard electrode, must be known.

It is necessary to define a solution of a certain concentration, temperature, speed of voltage supply, i.e., scanning speed (1 mV/s is used most commonly), equivalent weight, and density of the disc [27]. Two more elements are set: the initial and final potential. During the experiment, the potential includes the cathodic and anodic branches.

#### 2.2.2. Cluster Analyses and Principal Component Analyses

Cluster analysis (CA) and principal component analysis (PCA) are mathematical methods that are applied most frequently in natural sciences for the qualification and classification of large amounts of experimental data of different origins and for the determination of relationships between data [24,25,28,29,30,31].

Cluster analysis is a multivariate method that classifies the analysed data in clusters, based on the similarity between the data and particular parameters. Most scientific studies apply agglomerative hierarchical clustering. This type of cluster analysis compares the analysed data, and creates clusters of data that are similar according to predetermined criteria. There are different degrees of similarities and differences that can be used in data classification [32,33].

Principal component analysis is a multivariate method that is applicable to studies that contain huge amounts of heterogenous data or data obtained in different ways. The main advantage of this analysis is the recognition and exclusion of the redundant data that do not contain new information and encumber an analysed system. The scope of the analysed data is, thus, reduced without significant losses of important information. PCA identifies new variables, i.e., principal components (PC), which represent linear combinations of the original variables. Unlike original variables, principal components are not interrelated. The first principal component (PC1) exhibits the maximum variance of the analysed data set, while each subsequent principal component represents the highest following residual variance. The number of principal components that emerge in PCA equals the number of variables in a system. However, the number of principal components, which provide an adequate description of an analysed matrix can be lower than the number of original variables if there are significant correlations between the analysed data [34,35,36].

For the purposes of this paper, both multivariate analyses were conducted on a matrix in order to define the qualitative and quantitative impacts of certain types of marine environments on the corrosive behaviour of the NiTi discs. In the matrixused, experimentally obtained corrosion parameters from the EDX analysis were variables (columns), while different measuring points on the NiTi discs (spectrums) represented rows.

Before the application of a particular multivariate analysis, the data from the matrix were standardised, in order to ensure an equal influence of all the analysed factors. Euclidean distance was applied to all calculations as a measure of difference, while clusters were formed by means of Ward’s method. The multivariate analysis was conducted by means of Statistica v.13.5.017 software (StatSoft Inc., Tulsa, OK, USA).

### 2.3. Data Collecting Analysis

The experiment with the NiTi discs was conducted in three different seawater locations between August 2018 and May 2019. Samples in seawater were influenced by seawater between August 2018 and March 2019, while samples in the atmosphere and tidal zone were influenced by the atmosphere and tidal zone from December 2018 to May 2019. An adequate evaluation of the influences of the seawater environment on corrosion is required to conclude relevant environment parameters for the Bay of Kotor and was observed over a long period of time prior to the research.

The average temperatures of the sea and atmosphere were below 20 °C during the period of observation, while the average monthly temperatures of the sea were higher than the average temperatures of the air. The difference between temperature values varied between 0.8 °C for August and 10.3 °C for December [33]. Likewise, the maximum temperatures of the sea were considerably lower than the maximum monthly air temperatures that varied between 0.2 °C in December and 10.9 °C in March. The minimum sea temperatures, on the other hand, were significantly higher than the minimum air temperatures that varied between 12.0 °C in October and 22.6 °C in December. This indicates notably lower aberrations in the sea temperature in comparison with the air temperature [37].

The data about seawater temperature, conductivity, and salinity show that there were no significant aberrations in the values obtained on the sea depth up to 5 m. The average temperature was 18 °C, conductivity was between 44.29 and 47.45, and salinity between 30.83 and 28.79 [37].

The salinity and conductivity of the sea decreases on the surface between September and May, due to the rainy season and the inflow of fresh water. Compared to the atmosphere’s collected temperature data, higher sea temperatures in the period observed and other influences of the sea (salinity and conductivity) rendered the corrosion processes in the sea significantly faster than the corrosion processes in the atmosphere.

## 3. Results

Considering the conceptual model shown in Figure 2, in the following two subsectionsthe results in two specific directions of research will be shown. The first is corrosion depth and the second is changes in the chemical composition of the NiTi discs due to different environment and exposure time influence.

A macro view of the discs in three different seawater environments and the corresponding optical micrograph are presented in the Figure 3, Figure 4 and Figure 5.

Considering the macro view, it can be concluded that the disc sample in the atmosphere did not have any surface changes while the sample in the tidal zone has visible fouling on the sample surfaces. That is due to intensive flora and fauna on sea surface. Micro view presents intensive changes in samples exposed to sea water compared to the tidal zone, and minimal changes on samples in the atmosphere.

### 3.1. Comparative Results

Before starting the polarisation measurement, the system should be stabilised, i.e., after immersion in the electrolyte, the electrical circuit between the working and auxiliary electrodes is left open and the potential difference between the reference and working electrodes is monitored as a function of time. After a certain time, an approximately steady state is established at some potential value, which is equal to the corrosion potential of e_corr_. A steady state is established on the surface of the disc immersed in the electrolyte, the anodic metal dissolution current and the cathodic reduction current are of the same value, but of the opposite direction. By monitoring the changes in the stationary potential of the open circuit over time, we obtained data on the corrosion behaviour of the tested material in that environment. The values are shown in Table 1, and the diagram in Figure 6 shows a shift in the potential of the tested discs towards more positive values. A steady state was established after a time of 400 s. It is important to note that the most positive values of the potential were observed in the discs in the atmosphere, which is explained by the lower corrosion rate of the disc in the tested solution. The initial disc had the most negative potential and highest corrosion rate.

The shift of the potential of alloys towards more positive values is explained by passivation, i.e., the formation of an oxide film on the surface of the tested discs. The protective layer has an increasing thickness and becomes more compact over time. This film prevents the passage of aggressive chloride ions from the solution, thus protecting the material from further corrosion.

Linear polarisation is an electrochemical technique of determining the corrosion rate based on the determination of the corrosion current from the slope of the polarisation curve in the immediate vicinity of the corrosion potential, as well as on the determination of the polarisation resistance, Rp. Rp is defined as the slope of the polarisation curve at the corrosion potential. The current-potential ratio in the vicinity of the open-circuit potential (±20 mV) is monitored experimentally by the linear polarisation method. The values of the polarisation resistance of the corrosion current density and the potential when the current density is equal to zero, i.e., E (j = 0) were determined by extrapolating the linear dependences from the Figure 7. The values are shown in Table 2. Based on the results, it can be concluded that the NiTi discs were stable in the tested solution, i.e., they have a low corrosion rate. Comparing the results based on the value of the polarisation resistance Rp and the most positive potential when the current density is equal to zero, the disc that had been in the atmosphere had the lowest corrosion rate. The highest corrosion rate was with the disc in seawater because this disc had the highest value of current density and lowest value of polarisation resistance. The initial disc had the most negative potential, which corresponds to the highest corrosion rate. Polarisation resistance i.e., corrosion current density are the most important and relevant parameters for determining a corrosion rate. OCP potential does not provide exact values of corrosion rate, as do polarisation resistance and corrosion current density.

The potentiodynamic method is based on the Butler-Volmer equation, which describes the total current passing through the phase boundary at which one anodic and one cathodic reaction occur, which are not under diffusion control. Figure 8 shows the potentiodynamic cathode and anode polarisation curves of the tested discsin 3% NaCl solution. The current density was recorded in the potential range relative to the open circuit potential (±200 mV), with a scanning speed of 1 mV/s. In both the cathode and anode regions, a small change in the current density with potential was observed, which indicates a low corrosion rate of the tested disc in the NaCl solution. By extrapolating the anode and cathode Tafel directions in their cross-section, we determined the values of the corrosion current density j_corr_ and the potential when the current density is equal to zero, i.e.,E (j = 0). Based on the values of j_corr_ and E (j = 0), which are given in Table 3, it can be concluded that the disc in seawater had the highest corrosion rate in the NaCl solution. As with the previous method, the highest corrosion rate was with the disc in seawater because this disc had the highest value of current density and lowest value of polarisation resistance. The initial disc had the most negative potential, which corresponds to the highest corrosion rate.

### 3.2. Results PCA and CA

#### Analysis of the EDX Results

Due to the complexity of the marine environment, the exposed discs can undergo different types of corrosion [38,39,40]. Other than electrochemical corrosion, which is the most frequent type of corrosion, stress corrosion and microbiological corrosion are also common [41].

After a six-month exposure of the NiTi discs to the influences of different types ofmarine environments, the surface morphologies and compositions of corrosion products were examined by means of SEM and EDX analysis. Figure 9a–c show the micrographs of the resulting effects on the surface of the NiTi discs in which the upper surface is shown. The corresponding metal content of the observed surfaces expressed with EDX analysis performed on all tested disks are shown in Figure 9d–f. The representation of areas where the EDX characterisations were performed are specially marked with Spectrum 1–7.

The SEM micrographs prove that all types of marine environments examined caused changes on the surfaces of the NiTi discs. Furthermore, the surface of the disc that was exposed to seawater for 6 months (Figure 9a) was covered unevenly by inorganic salts (the heterogeneous mixtures of Ca, Mg, K, Na, Cl^−^, and SO_4_^2−^) and the emerged corrosion products. This kind of non-uniform deposits cannot ensure long-term protection from corrosion [42]. Over the time interval examined, organic deposits (algae, microorganisms, etc.) and pitting corrosion were not detected on the surface [43,44]. The total content of metals (Figure 9d) also shows that the impact of seawater causes notable differences in the content of both metals, nickel and titanium, which was registered at particular measuring points (spectrums).

Figure 9b shows the impact of the ebb and flood tidal zone on the microstructure of the examined NiTi disc. As shown in Figure 9b, the deposits on the surface are more compact in comparison with the surface deposits that were submerged in seawater. There were no significant variations in the content of the metals; therefore, the corrosive impact of the two types of marine environments cannot be differentiated precisely.

On the other hand, in both types of marine environments, voids and cracks appeared on the surface layer, which confirms the high corrosion susceptibility of the NiTi discs examined in the marine environments [45].

The exposure of the discs to the atmosphere (Figure 9c) resulted in surface deposits that were thinner and more compact in comparison with the deposits that emerged in the seawater and tidal zone. Based on the EDX analysis, the content of the deposits on the examined surface included corrosion products and inorganic salts (airborne salts). Inorganic salts reached the surface because of the high relative humidity (RH) level of the marine atmosphere [46] and through the vapour that was condensed or absorbed by the exposed surfaces. Consequently, an electrolyte layer was formed on the surface of the discs [43,46,47]. Inorganic salts in the surface water film ensure the conductivity of the film. The conductivity of the film controls ion migration to metal surfaces directly and the subsequent corrosion degree caused by the ion migration [48].

As previously indicated, Figure 9d–f clearly show that there is a significant difference in the content of metals (nickel and titanium) that was registered at particular measuring points (spectrums) in all types of corrosive environments. In such cases, it is not easy to assess precisely the environmental impact and identify the differences in the effects of different types of marine environments on the corrosive behaviour of the examined discs. Therefore, the study relies on multivariate analyses (CA and PCA) in order to detect the differences between the effects of the particular types of marine environments on the corrosive behaviour of the examined NiTi discs. Based on the available data, the applied chemometric methods are efficient in the processing of the data whose origin varies and in the analysis of highly heterogeneous data [24,25,49]. The applicability of chemometric methods is based on the detection of significant deviations of data and the exclusion of deviating data from further analysis, which facilitates the identification of the dominant effects of a particular type of environment on the corrosion process. Figure 10 shows a dendrogram obtained by means of cluster analysis.

As shown in Figure 10, the examined types of marine environment scan be classified in two distinct clusters based on the impact on the corrosion of the NiTi discs. In that sense, the first cluster contains the data from several measuring points (spectrums) that show the effect of seawater (M) and several points that were under tidal influences (P). There was a notable difference between the data (vertical deviation). The data obtained from the measuring of the influences of seawater were particularly different from other data in the same cluster. In that regard, the data on seawater effects form a subcluster, with the exception of points M44, M46, and M42. These points represent untypical impacts of the environment, so the related data could be excluded from further analysis, with the aim of the identification of typical environmental effects and the differences and similarities between the corrosive effects.

The second cluster from the obtained dendrogram contains the remaining measured data, which indicates that the analysed types of environments do not exhibit great or notable differences interms of corrosive impact on the NiTi discs. However, within the second cluster, there are subclusters with clearly separated data based on the type of environment (the atmosphere, tidalzone, and seawater). Such findings indicate that cluster analysis can detect small differences between similar corrosive influences of the analysed types of environments during the period of exposure. Figure 11 shows the distribution of the different types of corrosive environments based on PCA.

As is notable in Figure 11, the first two principal components (PC1–PC2) lead to a similar separation of the analysed environment types, based on cluster analysis and the corrosive impact on the NiTi discs. There is an evident clustering and separation of the influences of the ebb and flood tidal zone (P) from the influences of seawater (M) and the atmosphere (V). The corrosive behaviour of the NiTi disc in the atmosphere was described by positive PC1 values and negative PC2 values. The corrosive behaviour affected by the tides was, besides positive PC1 values, also described by negative PC2 values. The least notable clustering was observed among the data corresponding to the influences of sea- water. Namely, the impact of seawater is different from the tidal impact in terms of the PC2 values, while there were no differences in relation to the impact of the atmosphere. The PC1 values corresponding to the influences of seawater do not exhibit a clear distinction; the PC1 values were both positive and negative. On the other hand, it is evident that the corrosion induced by the atmosphere produced a highly homogeneous surface structure, while the corrosion induced by the seawater created the most heterogeneous surface. Regardless of the surface structure, the corrosion mechanism of the NiTi disc is based on the formation of the oxides of titanium and nickel in all types of marine environments. This finding is substantiated by the comparison of the total content of metals (the sum of the titanium and nickel amounts) at all measuring points, with the total content of oxides. The comparison resulted in a linear dependence (Figure 12 and Equation (1)), which confirmed the statement that the degradation of metals within the examined alloy happens mostly through oxide formation.
All metals (w/w%) = 100.71–1.48 O (w/w%) (R = −0.970 sd = 7.13 N = 52 *p* < 0.0001) (1)

PCA forms principal components as a linear combination of the original variables, and recognises and excludes redundant data that do not contain new information and encumber the analysed system. For these reasons, PC1, as the first principal component, should describe the corrosion of the analysed NiTi disc prevalently in all types of environments. The obtained PC1 values for all types of environments were correlated with the oxygen content obtained by means of the EDX analysis (Figure 13) in order to confirm the previous statement.

According to Figure 13, the values of PC1 are changing linearly in the function of the oxygen content that was obtained by the EDX analysis. There is a clear difference in the corrosive behaviour of the analysed discs (PC1) depending on the type of marine environment (Equations (2)–(4)), which was not the case in the linear regression analysis (Figure 12).

Seawater
PC1 = 1.833−0.084 O_2_ (w/w%) (R = −0.932 sd = 0.553 N = 22 *p*< 0.0001)(2)

Tidal zone
PC1 = 2.625−0.065 O_2_ (w/w%) (R = −0.961 sd = 0.345 N = 16 *p*< 0.0001)(3)

Atmosphere
PC1 = 2.223−0.041 O_2_ (w/w%) (R = −0.974 sd = 0.049 N = 14 *p*< 0.0001)(4)

The obtained results confirm that the PCA method can detect subtle quantitative differences between the corrosive effects of different types of marine environments, even though the studied corrosive effects are quite similar. The chemometric methods used (CA and PCA) are, therefore, sensitive enough to determine the existence of minimal differences between the corrosive effects of the environment depending on the type of material selected and analysed. This was confirmed on the NiTi discs. The use of chemometric methods ensures the identification and differentiation between the effects of different types of environments on the corrosion resistance or degradation of NiTi discs under certain conditions.

## 4. Conclusions

The main findings are the following:-Comparing the results, the NiTi disc that had been in the atmosphere had the lowest corrosion rate because of the most positive values of the corrosion potential and highest value of polarisation resistance.-The highest corrosion rate was with the NiTi disc in seawater because this disc had the highest value of current density and the initial disc had the most negative potential.-All types of marine environments caused changes on the surfaces of the NiTi discs, whereby there was a notable difference in the content of metals at particular measuring points (spectrums).-Regardless of the type of environment, the corrosion of the analysed NiTi discs happens through oxide formation. The obtained results confirm that PCA can detect subtle quantitative differences among the corrosive influences of the types of marine environments, although the examined corrosive influences are quite similar. The applied chemometric methods (CA and PCA) are, therefore, sensitive enough to register the existence of slight differences among corrosive environmental influences on the NiTi SMA analysed.-The value of the first principal component (PC1) describes the amount of the formed oxides quantitatively. Oxide amounts might vary, depending on the type of corrosive environment to which the alloy was exposed.

## Figures and Tables

**Figure 1 materials-15-02841-f001:**
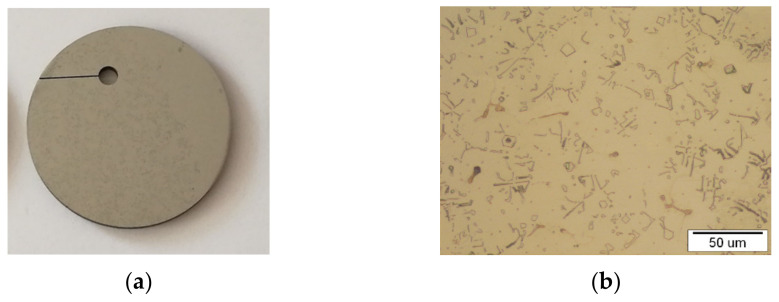
NiTi disc: (**a**) presentation of a macro view and (**b**) representative optical microstructure.

**Figure 2 materials-15-02841-f002:**
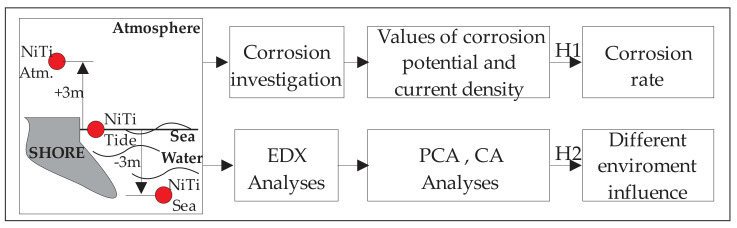
The scheme of the conceptual model of the research for the NiTidiscs.

**Figure 3 materials-15-02841-f003:**
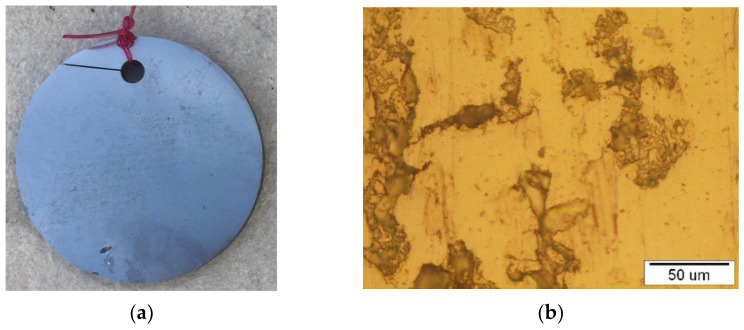
NiTi disc after 6 months in atmosphere: (**a**) presentation of the macro view and (**b**) representative optical microstructure.

**Figure 4 materials-15-02841-f004:**
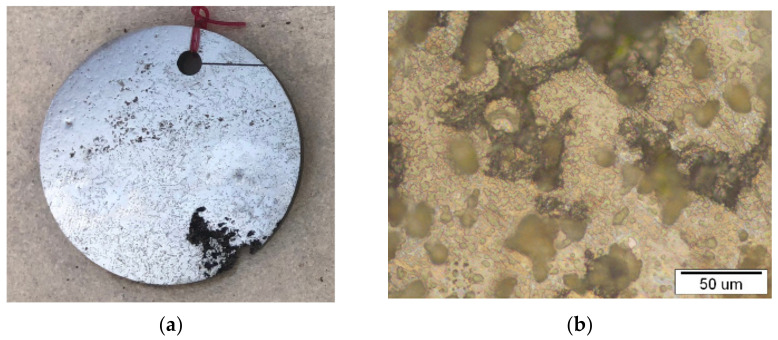
NiTi disc after 6 months in a tidal zone: (**a**) presentation of a macro view and (**b**) representative optical microstructure.

**Figure 5 materials-15-02841-f005:**
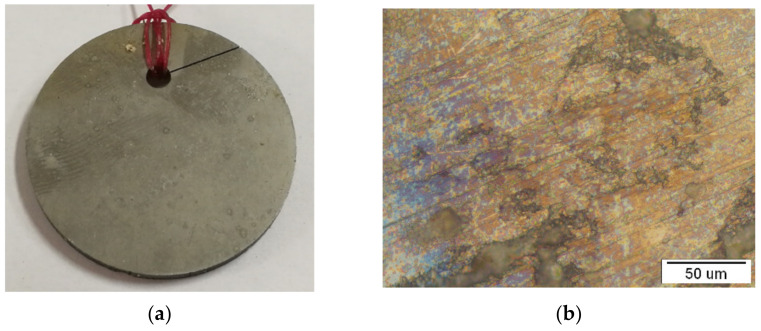
NiTi disc after 6 months in seawater: (**a**) presentation of a macro view and (**b**) representative optical microstructure.

**Figure 6 materials-15-02841-f006:**
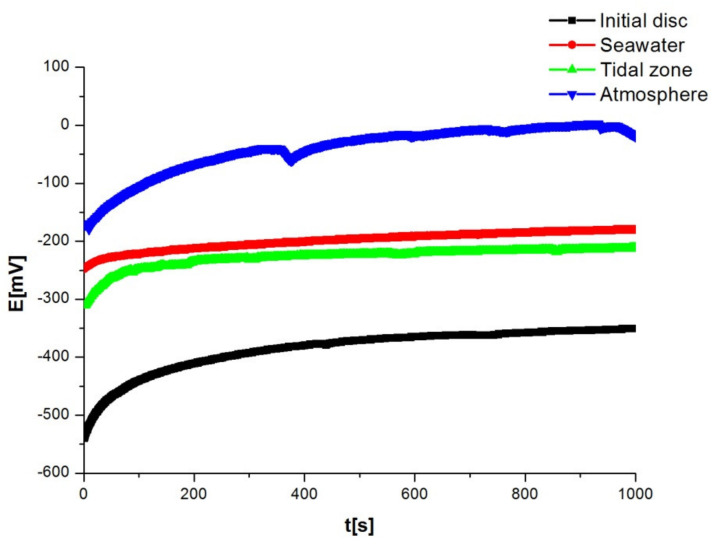
Results obtained by changing the corrosion potential as a function of time.

**Figure 7 materials-15-02841-f007:**
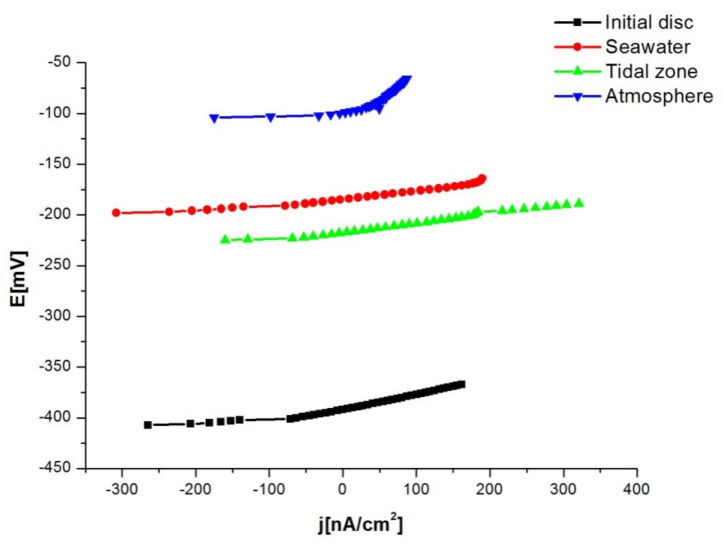
Results obtained through the polarisation resistance method (linear polarisation).

**Figure 8 materials-15-02841-f008:**
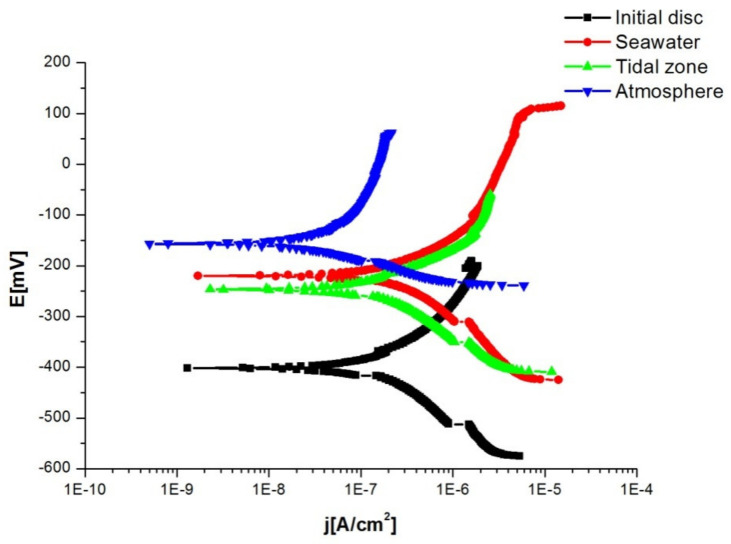
Results obtained throughthe potentiodynamic method.

**Figure 9 materials-15-02841-f009:**
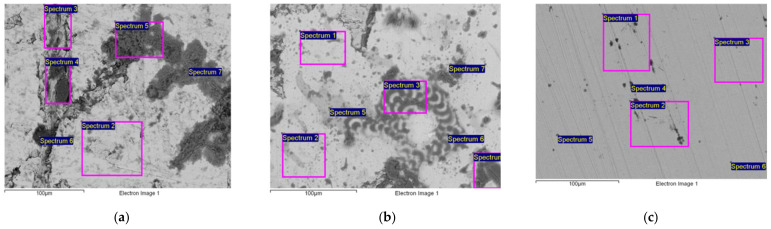
The representative micrographs of the NiTi discs with the corresponding metal content results after 6 months of exposure: (**a**,**d**) in seawater; (**b**,**e**) in a tidal zone; (**c**,**f**) in atmosphere.

**Figure 10 materials-15-02841-f010:**
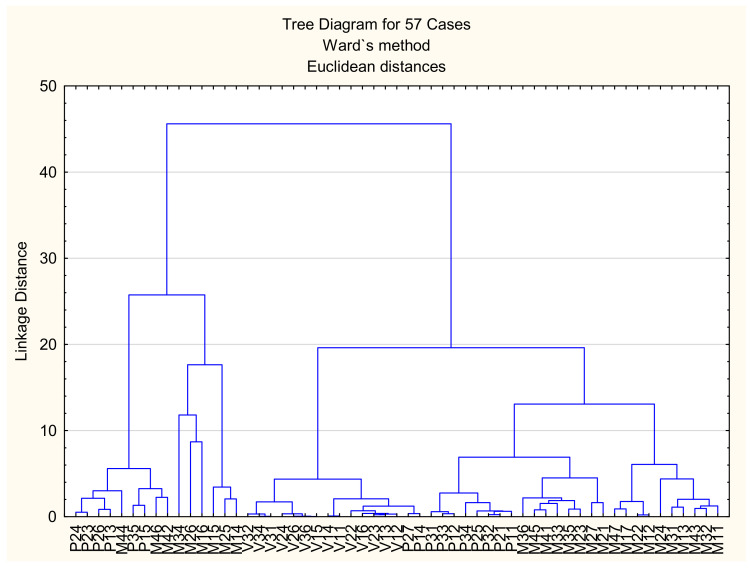
The dendrogram of the clustering of the examined types of marine environments based on the results of the EDX analysis.

**Figure 11 materials-15-02841-f011:**
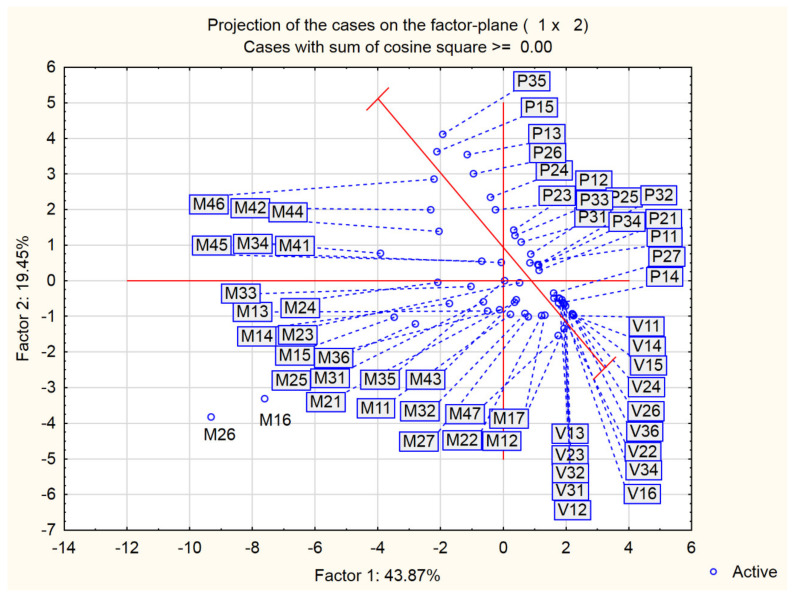
The score plot of the types of marine environments examined (PC1-PC2).

**Figure 12 materials-15-02841-f012:**
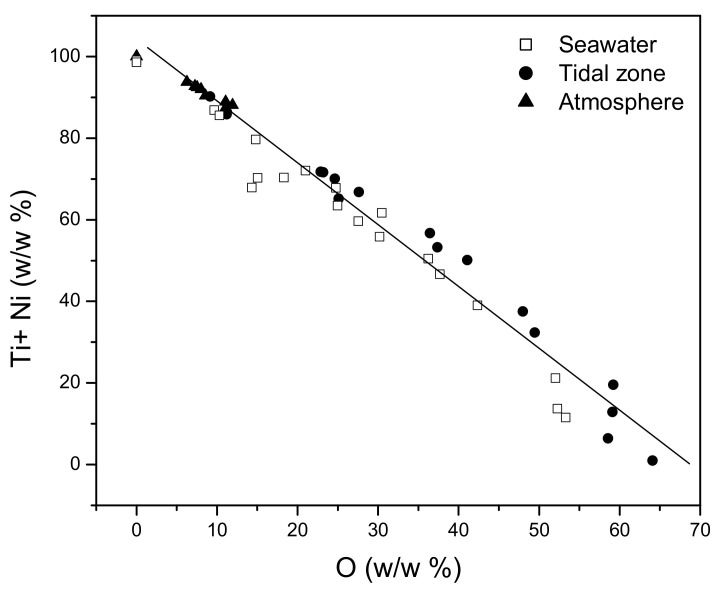
The change in the total content of metals in the function of oxygen content.

**Figure 13 materials-15-02841-f013:**
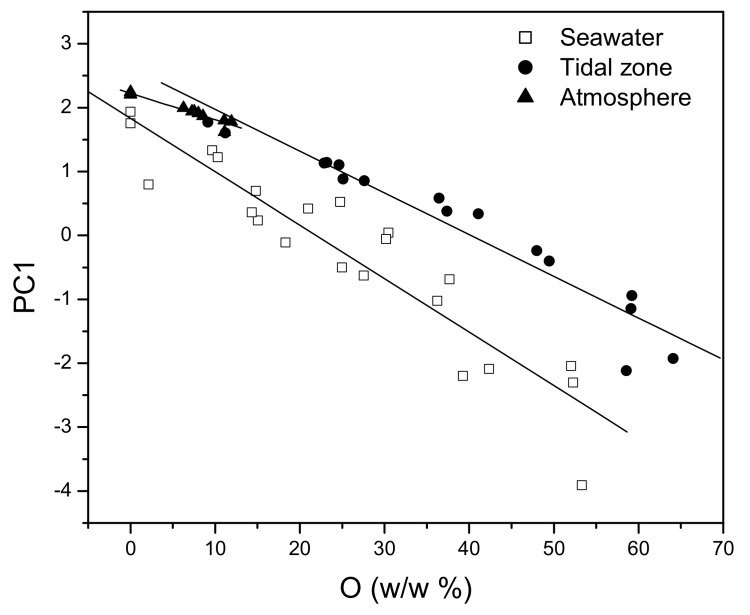
The change in PC1 values in the function of oxygen content.

**Table 1 materials-15-02841-t001:** Results obtained by changing the corrosion potential as a function of time.

Disc	e_initial_ [mV]	e_final_ [mV]
Initial disc	−540	−351
Seawater	−248	−180
Tidal zone	−311	−212
Atmosphere	−172	−19

**Table 2 materials-15-02841-t002:** Results obtained through the polarisation resistance method (linear polarisation).

Disc	E (j = 0) [mV]	Rp [kΩ]	j_corr_ [nA/cm^2^]
Initial disc	−391.5	144.4	150.4
Seawater	−182.8	70.11	309.7
Tidal zone	−217.4	96.86	224.2
Atmosphere	−106.6	416.0	52.20

**Table 3 materials-15-02841-t003:** Results obtained through the potentiodynamic method.

Disc	OCP [mV]	E (j = 0) [mV]	b_k_ [mV/dec]	b_a_ [mV/dec]	j_corr_ [nA/cm^2^]
Initial disc	−375	−401.1	154.9	193.3	227.9
Seawater	−225	−219.8	279.6	219.8	609.8
Tidal zone	−209	−246.6	170.8	151.7	281.5
Atmosphere	−39	−158.3	378.1	65.36	58.24

## Data Availability

Not applicable.

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
