# Peer review of "Corrosion of NiTiDiscs in Different Seawater Environments"

_materials, 2022, doi:10.3390/ma15082841_

Round 1

Reviewer 1 Report

Reviewer Recommendation and Comments for manuscript materials-1650153 with the title: “Corrosion of NiTi discs in different seawater environments”, authors: J. Pješčić-Šćepanović, G. Vastag, Š. Ivošević, N. Kovač, R. Rudolf.

The authors present the corrosion analysis and changes in the chemical composition of NiTi alloy in seawater. The chemical composition of the NiTi was measured by Inductively Coupled Plasma and X-Ray Fluorescence before the experiment and by semiquantitative anlysis after 6 months of exposure.

The article may be published after revision.

The main comments that I find useful for improving the quality of the article are presented below:

*General remark. 2.1. Materials. This part of the manuscript should contain more experimental details on obtaining NiTi by casting and rolling.

*line 160. Table 1. NiTi alloy (50% / 50%) was obtained using pure metals. The authors should explain the difference of 5.5% in the composition of the alloy.

*lines 160 and 165. Table 1 and Table 2. The data presented in these tables represent the subject of previous studies already published, according to [20]. With the consent of the authors, these tables must be deleted.

*line 222. What does it mean?: ”During the experiment, the potential usually shifts from the cathode to the anode one.”

*General remark. The data shown in Figures 3-5 should be commented on in more detail.

*General remark. e for potential should be replaced with E.

*General remark. 3.1. Comparative results. It is not clear how the electrochemical parameters for the atmospheric sample were obtained. How were OCP, Jcorr, etc. determined? in the gaseous atmosphere?

*line 358. ”Results PCA i CA” it means ”Results PCA and CA”

*line 374. (c) and (f) in seawater. (atmosphere)

*The typos must be corrected.

-line 55. ”The cost of NiTi is high”. Line 56. ”The fabrication costs of NiTi were low”

-line 57. Theywere

-line 72. which, finally,

-line 144. ml

-line 210. i.e. e / E

-line 215. The e / E

etc.

*The Materials journal require a specific format of references, authors must pay more attention in their writing.

*There are some grammar and typing mistakes.

*The authors must revise the entire manuscript.

Author Response

Please find our answers.

Reviewer 2 Report

The submission is devoted to study of NITi corrosion in real seawater condition during 6 month. The research topic is very interesting and has practical importance. The study is quite complete and has interesting results. It is especially worth noting that the authors actively used chemometrics to analyze the results.

Despite this, the manuscript has many disadvantages.

  1. There is little specificity in the abstract and no specific results and conclusions are described.
  2. The ultimate goal of the work is not entirely clear. It is also not clear what is the benefit and importance of the results obtained for science and industry? What is the novelty and significance of the study?
  3. The introduction is too big. There is a lot of redundant information about alloys with shape memory and history. However, there is no data about specific application of nitinol in the Maritime industry.
  4. Lines 100-101. …that occurs through the interaction between the metal 100 surface, the seawater and the conductivity of the seawater. The meaning of the sentence is not entirely clear. Please clarify
  5. Lines 120-121. A reference to the work in question is required.
  6. The materials and methods show the results of the work (fig 1, table 1-2). This is not correct in terms of the structure of the article.
  7. Tables and figures should go immediately after the text referring to them. Without this, it is very difficult to read the text. There is also no discussion of Figures 3-5 in the text. There is no reference in the text to Table 3.
  8. Section 2.3. the change in water temperature and salinity is described in detail, but it is not indicated in what time period the measurement was carried out, i.e. from what to what month.
  9. A disk “atmosphere” has the greatest resistance, even better than the initial one. Why? Is there a stronger passivation?
  10. Lines 335-338. “The highest corrosion rate was in the disc in seawater, because this disc had the highest value of current density and lowest value of polarisation resistance. The initial disc had the most negative potential which corresponds to the highest corrosion rate.” These two sentences contradict each other. It is necessary to explain in more detail how polarization resistance and potential affect real corrosion and which of the parameters is most important and relevant.
  11. Line 352-353 "Based on the values of jcorr and e (j=0), which are given in Table 5, it can be concluded that the disc in seawater had a lower corrosion rate in the NaCl solution". This is not so, the smallest potential and jcorr for the sample “atmosphere”
  12. On Fig 6d-f it is difficult to see anything. It is also desirable to provide EDX data on the initial nitinol
  13. What is the principle of area selection for EDX? Judging by Figure 9, is this not a completely random principle? And if this is not a random principle, then how correct are the results of CA and PCA?
  14. There is a lack of specifics when discussing the results of CA and PCA. What concrete conclusions can be drawn from these data?
  15. Conclusion. “Regardless of the type of environment, the corrosion of the analysed NiTi discs happens through oxide formation”. Indeed, oxidation occurs, but what about the dissolution of the material and the loss of mass. What was the weight loss of samples over 6 months?
  16. Conclusion. “The clustering of the results based on the type of corrosive environment can be conducted by the application of multivariate methods (CA and PCA) to the data obtained by means of the EDX analysis.” I wouldn't call it a conclusion. it is too general and without specifics
  17. Line 144. H2. Probably typo.
  18. Captions of figure 9. Please correct.

In conclusion, I would like to note that the research has interesting and important results, but the manuscript should to be remarkably improved, completed, and revised. Therefore, I would like to recommend this manuscript for publication in the Materials after major revision.

Author Response

Please find our answers.

Round 2

Reviewer 1 Report

Reviewer Recommendation and Comments for manuscript materials-1650153 with the title: “Corrosion of NiTi discs in different seawater environments”, authors: J. Pješčić-Šćepanović, G. Vastag, Š. Ivošević, N. Kovač, R. Rudolf.

The authors responded to the reviewers' requests by providing appropriate explanations.

Congratulations to the authors on their work, but there seems to be a slight rush. The manuscript also contains minor typos that need to be corrected carefully.

*General remark. e for potential should be replaced with E. Please see Tables 1,2,3, lines 833 and 891, 891.

*line 842. dencity

*The Materials journal require a specific format of references, authors must pay more attention in their writing (eg. no./no.).

Reviewer 2 Report

Most of the questions were answered fully and substantiated. Necessary changes in the text were also made. I believe that the manuscript can be published in its present form.